# An Integrated Canonical and Non-Canonical Wnt Signaling Network Controls Early Anterior–Posterior Axis Formation in Sea Urchin Embryos

**DOI:** 10.3390/jdb13040036

**Published:** 2025-10-08

**Authors:** Jennifer L. Fenner, Boyuan Wang, Cheikhouna Ka, Sujan Gautam, Ryan C. Range

**Affiliations:** Department of Biological Sciences, Auburn University, Auburn, AL 36849, USA; jlf0088@auburn.edu (J.L.F.); czk0033@auburn.edu (C.K.); gausuj777@gmail.com (S.G.)

**Keywords:** evolution, developmental biology, neuroectoderm, canonical Wnt signal transduction, non-canonical Wnt signaling, anterior–posterior, germ layers, gene regulatory networks

## Abstract

Wnt signaling is an ancient developmental mechanism that drives the initial specification and patterning of the primary axis in many metazoan embryos. Yet, it is unclear how exactly the various Wnt components interact in most Wnt-mediated developmental processes as well as in the molecular mechanism regulating adult tissue homeostasis. Recent work in invertebrate deuterostome sea urchin embryos indicates that three different Wnt signaling pathways (Wnt/β-catenin, Wnt/JNK, and Wnt/PKC) form an interconnected Wnt signaling network that specifies and patterns the primary anterior–posterior (AP) axis. Here, we detail our current knowledge of this critical regulatory process in sea urchin embryos. We also illustrate examples from a diverse group of metazoans, from cnidarians to vertebrates, that suggest aspects of the sea urchin AP Wnt signaling network are deeply conserved. We explore how the sea urchin is an excellent model to elucidate a detailed molecular understanding of AP axis specification and patterning that can be used for identifying unifying developmental principles across animals.

## 1. Introduction

Embryonic axes are the foundation upon which adult animal body plans are created, making their establishment during the early stages of embryogenesis one of the fundamental questions in evolutionary and developmental biology. The first embryonic axis is generally established during early cleavage in most metazoans, and it will define the head-to-tail primary axis (e.g., the anterior–posterior axis [AP] in bilaterians). Transiently active Wnt signaling gradients are the driving force behind the early specification and patterning of territories along the primary axis in most animals studied thus far [1,2]. Evolutionarily, Wnt signaling pathways are deeply homologous and are constructed from a complex set of extracellular modulators, receptors, co-receptors, as well as intracellular modulators and transcriptional effectors [3,4,5,6]. Over the course of the last four decades, a large body of work in several major model organisms (e.g., mouse, Drosophila, and *C. elegans*) has established how many of the components involved in these pathways are used to specify and pattern the axes [7,8,9]. Yet, there are still major gaps in our knowledge about how these components function and interact to drive axial formation. This gap is even more pronounced in invertebrate model organisms outside of Drosophila and *C. elegans*, preventing us from defining the similarities and differences among axial patterning mechanisms that would help us understand the evolution of animal body forms.

Beginning in the 1980s, several studies established that the AP Hox gene cluster originally identified in *Drosophila* also patterns the trunk regions along the primary AP axis in many bilaterian embryos. This remarkable conservation was the first, and arguably the most powerful, example of a conserved axial patterning mechanism that is shared among many bilaterians [10,11,12,13,14,15,16,17]. Subsequent work over the last three decades using progressively more sophisticated molecular techniques (e.g., RNAi, morpholinos, CRISPR) has made ‘non-genetic’ as well as ‘non-traditional’ model embryos from many phyla accessible for expression and functional analyses. Studies in these non-traditional model organisms have established that Wnt signaling gradients are another remarkably conserved primary axis patterning mechanism [1,2,3,6,18]. In the 1990s, studies established that the “canonical” Wnt/β-catenin (cWnt) pathway (Figure 1A) is critical for the specification of both the AP and DV axes in vertebrate *Xenopus* embryos [19,20]. Soon after, studies in mouse and invertebrate deuterostome sea urchin embryos were the next organisms to show that posterior-to-anterior cWnt signaling gradients are essential for specification and patterning germ layer territories along the primary axis, establishing this mechanism as a conserved feature in chordate and non-chordate deuterostome embryos [21,22,23,24,25,26]. These vertebrate and sea urchin studies led to similar observations in a diverse group of invertebrate embryos that strongly suggest that Wnt signaling gradients are an ancient molecular mechanism used to pattern the primary axis in many metazoans (reviewed in [1,5,18]). Perhaps one of the most evolutionarily important of these early studies is one from the Wikramanayake lab that found that a cWnt gradient is critical for the establishment of the primary axis in the cnidarian *Nematastella* [27]. This study showed for the first time that control of early primary axis formation by cWnt signaling gradients existed in pre-bilaterian animals. Current studies in Hydra have gone on to yield further support, establishing that the Wnt3/β-catenin/TCF/Sp5/Zic4 gene network regulates body axis formation by restricting head organizer activity [28,29]. Subsequent expression and functional studies in several invertebrate bilaterian and non-bilaterian embryos showed that a nested, overlapping pattern of Wnt ligands along the primary axis, termed the ‘Wnt code’ or ‘Wnt landscape’, is essential for specification and patterning their primary axes [6,30,31] (see Figure 2 for the sea urchin Wnt landscape during cleavage and blastula stages). Interestingly, this Wnt patterning mechanism initially appeared to predate the use of the Hox code for primary axis patterning, strongly suggesting that it is more ancient [30]. However, recent functional evidence from the Martindale lab suggests that Hox genes are necessary for specification/patterning of territories along the primary axis in the cnidarian *Nematostella* [32]. Additionally, in early branching metazoans, sponges and ctenophores, Hox genes were once thought to be absent, but work in calcareous sponges and *Mnemiopsis* revealed developmental expression of Hox and Parahox genes [33,34,35,36,37,38]. Thus, it is unclear which early primary axis patterning mechanism is more ancient, but it has been proposed that the evolution of the deeply conserved Wnt signaling pathway (along with the TGF- β signaling pathways) may have led to the first symmetry breaking events in early multicellular organisms [5,39].

Like most cell-to-cell signaling pathways, Wnt signaling is involved in a vast range of developmental processes, including transcriptional regulation, morphogenesis, and cell proliferation, to name a few [1,3,5,40,41,42,43]. It is interesting to speculate that this functional promiscuity is largely due to the inherently complex nature of Wnt signaling pathways. Both Wnt ligands and their Frizzled (Fzd) receptors belong to multigene families in most animal genomes, with vertebrates typically possessing the largest complement of ~19 Wnt ligands and 10 Fzd receptors, which could result in 190 possible pairings [41]. In addition to this pairwise complexity, there are also several Wnt co-receptors and secreted modulators (agonists and antagonists) that add to the staggering number of potential combinations of protein–protein interactions that could initiate Wnt signaling in a particular cellular context. Another level of complexity arises from the fact that any of these extracellular protein–protein interactions can potentially activate one or more intracellular signal transduction cascades [2,4,44]. Of these transduction mechanisms, the best characterized are the biochemical processes of the cWnt pathway that lead to β-catenin translocation to the nucleus and its subsequent interaction with the transcription factor TCF/LEF (Figure 1A) (reviewed in [42,43]). However, Wnt ligands can also activate intracellular signal transduction cascades that use different intracellular components, depending on the interactions among the various Wnt ligands, Fzd receptors, co-receptors, and secreted modulators. Two of the better characterized of these ‘non-canonical or alternative’ pathways are the Wnt/c-Jun N-terminal kinase (JNK) or Wnt/Planar Cell Polarity [PCP] and the Wnt/Ca^2+^ signaling pathways (reviewed in [4]), which use Ca^2+^ release and phosphorylated JNK as second messengers, respectively. Although both pathways are more often associated with their ability to influence cytoskeletal rearrangements, they have also been implicated in transcriptional regulation via transcriptional effectors other than β-catenin/TCF, such as Jun/ATF2 (Wnt/JNK) and NFAT (Wnt/Ca^2+^) (see Figure 1B,C for details) [45,46,47,48]. Finally, many studies have shown that two or more of these pathways are active simultaneously in the same cells or territories, sharing components at most levels of the cascade and often forming cross-regulatory interactions [2,4,44]. These data have led to the idea that Wnt proteins can activate complex signaling networks whose combined inputs are balanced to influence cellular behavior in a given context, further adding to the complexity of this signaling family.

**Figure 1 jdb-13-00036-f001:**
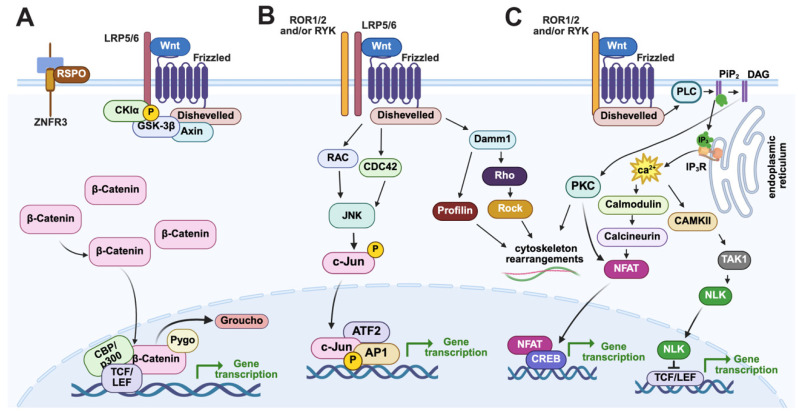
Graphical illustration of the canonical and non-canonical Wnt signaling pathways. (**A**) In the Wnt/β-catenin signaling pathway, Wnt1 ligands interact with Fzd receptors and the co-receptor LRP5/6, leading to the recruitment of disheveled to the membrane. Disheveled subsequently promotes a complex molecular mechanism that inhibits the activity of the β-catenin destruction complex (APC, Axin, CK1-alpha, and GSK3- β), allowing β-catenin to translocate to the nucleus, interact with the transcription factor TCF/LEF, and activate target gene transcription. (**B**,**C**) The non-canonical Wnt/JNK and Wnt/Ca^2+^ use Fzd receptors have been shown to interact with co-receptors (e.g., ROR and RYK) to activate different intracellular signaling cascades. (**B**) In the Wnt/JNK pathway, Wnt/Fzd interactions activate a signal transduction cascade, often through Dsh, that leads to cytoskeletal rearrangements and/or transcriptional activation via interactions with the transcriptional effectors ATF2/Jun. (**C**) Wnts interacting with Fzd receptors can activate a G-protein signaling pathway that triggers Ca^2+^ release from the endoplasmic reticulum, leading to activation of transduction factors like PKC and Calcineurin that can influence cytoskeletal reorganization and/or NFAT-mediated gene transcription. It is important to note three things: (1) Dsh proteins use distinct domains to activate the different Wnt signaling pathways (N-terminal DIX, the central PDZ, and the C-terminal DEP domains). (2) Wnt, RYK, and ROR interactions have been shown to regulate all three Wnt pathways. (3) Studies indicate that Wnt interaction with ROR and Ryk can activate non-canonical Wnt/JNK and Wnt/Ca^2+^ signaling independently of Frizzled receptors. Image created in BioRender (https://BioRender.com/8gmwkh3, accessed on 6 July 2025).

**Figure 2 jdb-13-00036-f002:**
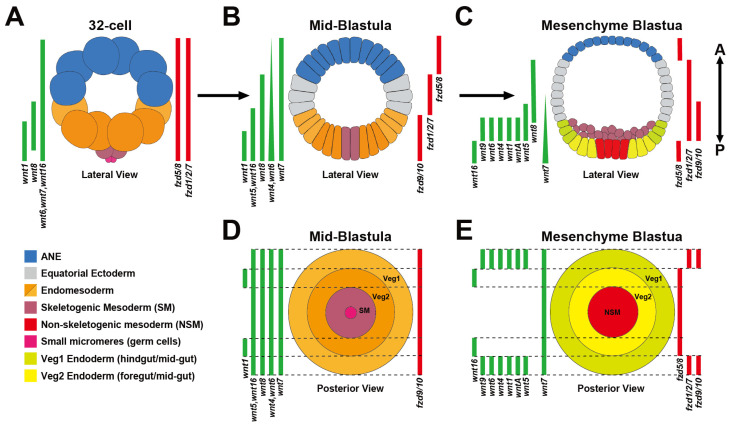
Wnt landscape gene expression during sea urchin primary germ layer GRN specification and patterning. (**A**) 32-cell stage embryos ubiquitously express three maternally supplied Wnt ligands (Wnt6, 7, and 16) and two Frizzled receptors (Fzd1/2/7 and Fzd5/8) that overlap with the zygotic expression of Wnt1 and Wnt8 in the endomesoderm territory. (**B**) Mid-blastula stage embryos show staggered arrays of 7 Wnt ligands (Wnt1, 4, 5, 6, 7, 8, and 16) and 3 Fzd receptors (Fzd1/2/7, Fzd5/8, and Fzd9/10) along the anterior–posterior axis. (**C**) At the beginning of gastrulation, dynamic expression changes result in staggered expression of 9 Wnt ligands (WntA, 1, 4, 5, 6, 7, 8, 9, and 16) along with Fzd1/2/7, Fzd5/8, and Fzd9/10 along the anterior–posterior axis. (**D**) Of the Wnt genes expressed in mid-blastulae, all are expressed throughout the entire endoderm and mesoderm territory except for *wnt1*, while Fzd9/10 is the only Fzd receptor expressed in these territories. (**E**) By the mesenchyme blastula stage, 8 of 11 Wnt genes, as well as Fzd1/2/7, Fzd5/8, and Fzd9/10, have resolved to their final expression domains within the posterior endoderm and mesoderm territories prior to gastrulation. Data taken from [49,50,51,52,53]. Lateral embryo images in the figure created in BioRender (https://BioRender.com/8gmwkh3, accessed on 6 July 2025).

In this review, we discuss the expression and functional data that show how the three Wnt pathways described above form an integrated Wnt signaling network that governs primary AP axis formation in sea urchin embryos. Taken together, the data presented here challenge the widely accepted view that a posterior-to-anterior cWnt gradient is the main driver of early AP axis specification and patterning in the sea urchin. We will highlight expression and functional studies in several species, including vertebrates, that demonstrate that key aspects of the sea urchin AP Wnt signaling network likely existed in the common deuterostome ancestor among chordates and ambulacrarians. Finally, we hope to show that the relative morphological and genomic simplicity of sea urchin embryos in combination with their well characterized gene regulatory networks (GRNs) makes them an excellent model for elucidating the complex signaling interactions governing the earliest stages of animal development.

## 2. Dynamic Spatiotemporal Expression of the Early Sea Urchin AP Wnt Landscape

Sea urchins are echinoderm deuterostomes that are grouped with their sister phylum Hemichordata into the superphylum ambulacraria [54]. Because embryos from both groups are closely related to chordates, it has allowed researchers to identify many shared mechanisms for Wnt-mediated AP formation among deuterostomes as well as animals in general [2,55,56]. As we discuss below, the embryonic sea urchin primary axis is established during oogenesis [57], and like many animals, this axis extends from the animal pole (determined by the site of polar body budding from the oocyte) to the opposite vegetal pole where the blastopore will form (termed the animal–vegetal [AV] axis) [58,59] (reviewed in [56,60]). Vegetally localized canonical cWnt signaling initiates the specification and patterning of the primary germ layers (endoderm, mesoderm, and ectoderm) along the AV axis starting around the 16-cell stage [21,23,57,61,62]. One cell division later, the first evidence for zygotic control of embryonic patterning becomes evident when several early components of the endomesoderm (EM) GRN (e.g., *wnt1*, *wnt8*, *blimp1b*, and *eve*) and the two cardinal regulators of the anterior neuroectoderm (ANE) GRN, *six3/6* and *foxQ2*, are expressed in separate vegetal and animal blastomeres, respectively (Figure 2A) [27,49,50,63,64,65,66]. Beginning around the 60-cell stage, the ANE GRN components begin to be progressively downregulated in the central, equatorial ectoderm (Figure 2A), effectively restricting the ANE GRN to a discoidal territory around the anterior pole by the beginning of gastrulation at the mesenchyme blastula stage [50,63] (Figure 2A). We term this developmental process ‘ANE restriction’, and once it is complete, the exact positions of the germ layer GRNs are established along the AV axis [2,50,53,67] (Figure 2B). These GRNs include the endoderm and mesoderm GRNs, positioned around the posterior/vegetal pole; the equatorial ectodermal territory that will form ventral and dorsal ectodermal structures separated by the ciliary band and associated nerves; and the ANE GRN around the anterior/animal pole, which will form a centralized larval sensory/neurosecretory apical organ and surrounding neural tissues. As we discuss below, expression studies in many metazoans, including vertebrates, show that orthologues of many of these key sea urchin endomesoderm and ANE GRN components are expressed in the corresponding germ layer territories that are aligned along the early AP axis in these animals. And functional studies show that many of these orthologous genes play remarkably similar roles in these territories, as observed in sea urchin embryos [66,68,69]. Finally, it is a well-established fact that a gradient of Wnt signaling is essential to specify and pattern these conserved germ layer GRNs along the AP axis in these animals [2,6,30]. For these reasons we argue that these similarities indicate that the early sea urchin AV axis established during oogenesis prefigures the larval AP axis.

The diversification of Wnt ligands occurred before the split of bilaterians and non-bilaterians, with the ancestral eumetazoan genome thought to have contained 13 Wnts (Wnts1 through 11, WntA, and Wnt16) [6,31,40,70,71,72]. Sea urchin genomes contain almost a full complement of orthologues to the ancestral Wnt ligands [73,74], lacking only Wnt11 in all species and Wnt2 depending on the species [51]. Nine of these *wnt* genes are dynamically expressed during the cleavage and blastula stages when the germ layers are initially specified and patterned along the early AP/AV axis. Early on during the first 4 cleavages, maternally supplied Wnts (Wnt6, Wnt7, and Wnt16) dominate the Wnt landscape [51,53,74,75] (Figure 2A). Then, around the 16- to 32-cell stage, zygotic Wnt ligand expression begins with the activation of *wnt1* and *wnt8* transcripts. Both ligands are expressed in the early endomesoderm around the posterior/vegetal pole, with *wnt1* transcripts expressed in the skeletogenic mesoderm and endomesoderm cells and *wnt8* transcripts expressed in the endomesoderm territory [27,50,65] (Figure 2A). As embryogenesis progresses into the early blastula stages, *wnt1*, *wnt4*, *wnt5*, and *wnt16* are activated throughout the endomesoderm. The broadly expressed maternal *wnt16* transcripts are downregulated in the ectoderm, while *wnt8* expression expands anteriorly from the endomesoderm into the equatorial ectoderm (Figure 2B). The Wnt signaling landscape changes dynamically up until the primary germ layer positions are established by the beginning of gastrulation (mesenchyme blastula stage). During these late blastula stages, a nested pattern of wnt expression occurs within the endomesoderm when 7 Wnts are expressed in the anterior endoderm territory, dubbed veg1 (*wntA*, *1*, *4*, *5*, *6*, *7*, *9*), and 2 Wnts are expressed in the more posterior endoderm territory, dubbed veg2 (*wnt7* and *wnt16*) [51,52,53] (Figure 2C). It is important to note that Wnt7 (endoderm and ANE expression) and Wnt8 (equatorial ectoderm) are the only two Wnt ligands expressed outside the endomesoderm territory after the embryo transitions to zygotic control during late blastula stages [50,51,76] (Figure 2C).

The earliest diversification of Frizzled genes is also thought to have occurred before the split of non-bilaterians and bilaterians, containing an ancestral complement of four genes (Fzd1/2/7, Fzd4, Fzd5/8, and Fzd9/10) [6,77]. Like other invertebrate deuterostomes, including the basal chordate amphioxus, the sea urchin genome contains only these four Fzd orthologues, suggesting that the diversification of Fzd receptors in vertebrates occurred during the later stages of chordate evolution [2,51,78,79]. Of these ancestral Fzd genes, three are expressed during early AP specification and patterning in sea urchin embryos (*fzd1/2/7*, *fzd5/8*, and *fzd9/10*), and two, *fzd1/2/7* and *fzd5/8*, are maternally expressed throughout the embryo like maternally expressed *wnt6*, *wnt7*, and *wnt16* genes [50,51] (Figure 2A). This broad expression of these Wnt signaling components is notable because maternal Wnt6, Wnt16, Fzd5/8, and Fzd1/2/7 are critical for early AP patterning. In addition, their broad spatial expression suggests that they could be involved in other developmental processes taking place in any germ layer during cleavage and early blastula stages. The dynamic phase of *Fzd* gene expression begins around the 120-cell stage when maternal *fzd1/2/7* and *fzd5/8* transcripts are downregulated from around the posterior pole but remain expressed in the equatorial and ANE territories, respectively. At the same time, *fzd9/10* transcripts are initially activated in the posterior endomesoderm territory as *fzd1/2/7* and *fzd5/8* expression is downregulated there (Figure 2B) [50,51,67,80]. As the ANE restriction mechanism is reaching its terminal stages during the late blastula stages, *fzd5/8* transcripts begin to be activated in the non-skeletogenic mesoderm and veg2 endoderm territories, while *fzd1/2/7* and *fzd9/10* are expressed in more anterior veg1 endoderm and posterior ectoderm cells [51,81] (Figure 2C).

In addition to these traditional Wnt receptors, a recent spatiotemporal expression study showed that transcripts of two Wnt co-receptors, ROR1/2 and RYK, are also co-expressed in many of the same territories as the *Wnt* and *Fzd* genes throughout early anterior–posterior patterning. In multiple model systems, ROR1/2 and RYK have both been implemented as co-receptors for canonical and/or non-canonical Wnt/JNK or Wnt/Ca^2+^ signaling pathways depending on cellular context [82,83,84,85,86,87,88,89,90]. Both *ror1/2* and *ryk* transcripts in the sea urchin are broadly expressed throughout the embryo, overlapping with *wnts1*, *6*, *7*, *8,* and *16*, as well as *fzd1/2/7* and *fzd5/8* expression during cleavage stages (Figure 2A). Then zygotic *ror1/2* and *ryk* expression is dynamic, like *Wnt* and *Fzd* gene expression. By the mesenchyme blastula stage, *ror1/2* expression has been downregulated from the ANE territory and expressed primarily in the posterior endomesoderm and ventral ectoderm territories, whereas *ryk* is expressed in the ANE, ventral equatorial ectoderm, and endomesoderm territories [81]. Thus, there is a complex expression landscape of seven Wnt ligands overlapping with three Fzd receptors as well as ROR1/2 and RYK in the recently segregated endoderm and mesoderm at the beginning of gastrulation. Also, in the ectoderm, *wnt5*, *wnt7*, and *wnt8* are co-expressed with *fzd1/2/7* in the equatorial ectodermal region (Figure 2A). Together, these expression data suggest that ROR1/2 and RYK could play important roles in modulating which Wnt signaling pathways might be active in a particular territory.

A question that has confounded evolutionary developmental biologists for years is why there are so many Wnt ligands overlapping with just a few Fzd receptors, especially in the posterior endomesodermal territory? The remarkable deep evolutionary conservation of the 13 Wnt and 4 Fzd subfamilies suggests that these molecules are not performing redundant functions, even though their protein structures are similar. Over the last couple of decades, several studies have shown that the same Wnt and/or Fzd receptor can activate two or more different Wnt signaling branches, even in the same cells [4,91,92]. These studies have led to the idea that Wnt ligands and Fzd receptors are somewhat promiscuous. If this is the case, then it is possible that specific extracellular and/or intracellular modulators are necessary to determine the specific Wnt signaling branch that is activated in cells. Therefore, it will be important to establish which extracellular and/or intracellular modulators are involved in regulating specific Wnt signaling branches in any given context. The early sea urchin embryo offers a unique opportunity to explore these questions, since as we discuss below, there is currently no evidence for nuclear β-catenin outside the posterior endomesodermal territory in early sea urchin embryos [23,61]. And a recent temporal functional study showed that cWnt signaling is only active during cleavage and early blastula stages in these embryos [93]. Yet, two Fzd receptors and five Wnt ligands are expressed in anterior ectodermal cells, suggesting that they may activate non-canonical intracellular transduction cascades in these cells. Recent studies in this embryo are beginning to shed light on an integrated network of three different Wnt signaling pathways that often work in the same cells and territory.

### 2.1. Wnt/β-Catenin Signaling Positions the First Two Gene Regulatory Networks Along the AP Axis by the 32-Cell Stage

The first evidence of a zygotic patterning event in sea urchin embryos occurs with the activation of the EM GRN and ANE GRNs, respectively, in the posterior and anterior halves of the 32-cell embryo (Figure 3A). The sea urchin EM GRN is one of the best characterized of any deuterostome embryo [65,66,94,95] (Figure 3), and this regulatory network has provided a way to assess the evolution of endomesodermal regulatory programs across large phylogenetic distances [96,97,98,99,100]. Importantly, many of the genes in the sea urchin EM GRN, as well as the regulatory connections among them, are conserved among several metazoan models. For instance, orthologs of the sea urchin endoderm transcription factors Blimp1, Gata4/5/6, FoxA, and Otx (Figure 3A–D) form remarkably similar recursive regulatory interactions that are critical for endoderm specification in sea star embryos, which diverged from sea urchins ~550 mya [96,97]. Similar regulatory interactions have been observed in several metazoan embryos. For example, in protostome *C. elegans* embryos, Gata4/5/6 orthologues (End1 and End3) activate a FoxA orthologue (PHA-4), and this regulatory kernel initiates endoderm specification downstream of cWnt signaling as in sea urchins [101,102,103,104]. The regulatory interactions among the genes regulating endoderm specification downstream of cWnt are not well established in any chordate embryo. But it is interesting that many orthologs of the sea urchin/sea star endoderm regulatory sub-circuit are critical for endomesoderm specification downstream of posterior cWnt signaling in vertebrates, including Gata4/5/6, FoxA, and Blimp1/PDRM1 [105,106,107,108,109,110,111]. Based on these studies, it is tempting to speculate that key aspects of the endomesoderm regulatory program in sea urchins represent an ancestral metazoan developmental mechanism.

The sea urchin ANE GRN drives the formation of a centralized neurosensory structure called the apical organ (AO) around the anterior pole (Figure 4). Similar AOs form around the anterior poles of many invertebrate metazoan larvae, including chordate (amphioxus), ambulacrarian (echinoderm and hemichordate) (Figure 4B–D), and protostome larvae, as well as the aboral pole of cnidarian planulae [68,129,130]. Thus, this sensory structure provides a unifying developmental morphological module that can be compared across vast evolutionary distances. Functional studies in model species across these invertebrate phyla have shown that, as in sea urchins, Six3/6 and FoxQ2 sit near or at the top of the ANE GRNs that drive the formation of this neurosensory organ in many of these animals [64,131,132,133,134,135,136,137]. A recent systems level study found that many other ANE GRN components are shared broadly among deuterostomes and protostomes. In fact, they found that 28 out of the 31 identified ANE GRN components are also expressed within the ANE territories of many metazoans, including mice, zebrafish, *Xenopus*, *Drosophila*, and *C. elegans*, with 20 of these genes expressed in the ANEs of all these species [68]. Then, a subsequent study by Gattoni et al. (2025) [69] strengthened the idea that ANE GRNs are broadly conserved in invertebrate and vertebrate chordates. Using detailed single-cell data and multiplexed in situ hybridization assays, they spatially mapped most of the sea urchin ANE GRN to the developing retina and hypothalamus of the invertebrate chordate amphioxus as well as vertebrates (zebrafish). And they show that this conserved regulatory toolkit is restricted to the developing forebrain of amphioxus by posteriorizing Wnt signaling [69]. Importantly, functional studies in early vertebrate embryos show that Six3 sits near the top of the vertebrate ANE GRN (FoxQ2 has been lost in vertebrates) [132,138]. In addition, other functional studies show that many vertebrate ANE GRN genes (e.g., Dkk1/2/4, sFRP1/5, Pax6, Rx, EBF) shared with these other species are critical for the early development of the vertebrate forebrain and eye field (for review, see [2]). Thus, while vertebrate embryos lack apical organs, these similarities among ANE GRNs and their restriction to the anterior pole by posteriorizing Wnt signaling argue against the idea that these regulatory networks driving ANE formation evolved independently. Instead, these observations have led us and several other researchers to propose that the ANE GRN is an ancient regulatory ‘toolkit’ [2,64,139] and that aspects of this regulatory network are conserved among many metazoans with apical organs as well as chordates with more complex centralized anterior nervous systems.

A major question that has not been answered in the sea urchin, or for that matter in any other organism, is how the ANE GRN is initially activated. We know that cWnt signaling is critical to repress the ANE GRN in posterior blastomeres since the ANE GRN is expressed throughout most blastomeres in cWnt knockdown embryos as early as the 32-cell stage in sea urchins. Subsequently, these cWnt knockdown embryos develop into larvae that are almost entirely composed of anterior neuroectodermal neurons (e.g., serotonergic neurons) [50]. These results strongly suggest that a currently uncharacterized, broadly expressed, and likely maternally supplied regulatory mechanism activates the ANE GRN. The data supports the idea that posterior cWnt signaling activates an unknown EM GRN regulatory component(s) that represses zygotic ANE GRN expression in posterior blastomeres (Figure 2A and Figure 3A). This phenomenon has also been observed in several invertebrate embryos (cnidarians, hemichordates, and sea stars) in which cWnt signaling has been perturbed [79,140], suggesting that posterior Wnt signaling plays similar roles in these embryos. In vertebrates, posterior-to-anterior Wnt dependent AP patterning is most clearly observed during neuroectoderm formation. However, ventral BMP2, 4, and 7, as well as dorsal ADMP signaling, restrict the neuroectoderm territory to the dorsal side of these embryos during early AP axis formation [1]. Interestingly, experiments in *Xenopus* embryos showed that the neuroectoderm is expressed broadly if BMP2, 4, 7, and ADMP are all knocked down in the same embryo [141]. Using zebrafish, the Weinberg lab performed a similar study, but in addition to knocking down BMP2, 4, 7, and ADMP, they also knocked down both zebrafish β-catenin paralogues. ANE GRN components were even more broadly expressed in these embryos than in *Xenopus* embryos, which is remarkably similar to invertebrate cWnt knockdown phenotypes [142]. Finally, mouse and human embryonic stem cells raised in serum-free media that lacks exogenous morphogens (most importantly Wnt and BMP ligands) primarily develop into telencephalic (ANE) neural fates [143,144]. Collectively, these results argue that ANE GRNs in many animals may be activated by broadly active ‘default’ molecular mechanisms and that the role of early Wnt and BMP, as well as other morphogens, is to restrict this ANE potential to territories around the anterior pole.

### 2.2. A cWnt to Wnt/JNK Signaling Relay Mechanism Positions Ectodermal GRNs Along the AP Axis

The segregation of the endomesoderm and ANE GRNs by nβ-catenin in posterior/vegetal blastomeres at the 32-cell stage represents the first phase of AP patterning in sea urchin embryos (Figure 2A and Figure 3A). The second phase begins one cell division later when ANE GRN components, which initially only include *six3/6* and *foxq2*, are progressively downregulated in ectoderm cells around the equator of the embryo (Figure 3B,C). This second phase of the ANE restriction mechanism is responsible for creating the equatorial ectodermal territory as well as positioning the ANE GRN around the anterior pole [50,63,64]. As mentioned above, cWnt signaling downregulates ANE GRN components from posterior/vegetal blastomeres, but there is currently no evidence for either the presence of or direct function for nuclear β-catenin outside the posterior/vegetal endomesoderm territory. In fact, the data from several studies indicates that if cells have nuclearized β-catenin, they will be specified as endomesoderm and that cWnt signaling does not directly downregulate the ANE GRN in the equatorial ectoderm territory [23,50,63,75]. Instead, a recent series of epistasis experiments from our lab showed that a non-canonical Wnt signaling pathway drives this mechanism in the equatorial ectoderm [50,53,67,76,81,112]. The data support our current model that posterior cWnt signaling activates Wnt1 and Wnt8 beginning around the 16- to 32-cell stage in the posterior endomesoderm territory [21,27,49,57,65,145] (Figure 2A and Figure 3A). While both ligands are critical players in the EM GRN [27,49,65,95], data suggest that both ligands also diffuse into adjacent equatorial ectoderm, where they interact with the Fzd5/8 receptor, resulting in the phosphorylation of c-Jun Kinase (JNK) [50] (Figure 3B). Phospho-JNK then acts as a second messenger that activates the expression of the transcription factor Sp5, and morpholino knockdown of Sp5 shows that it is essential for the progressive downregulation of all ANE GRN components from the equatorial ectoderm during late cleavage and blastula stages [76]. Interestingly, Wnt1/Wnt8-Fzd5/8-JNK-Sp5 signaling also downregulates *fzd5/8* expression from the equatorial ectoderm in a classic negative feedback mechanism, resulting in the expression of *fzd5/8* around the anterior pole (Figure 3B–D). It is important to note that *wnt1* expression remains restricted to the endomesoderm territory during late cleavage and blastula stages, suggesting that it is likely only necessary for the initial stages of ANE restriction [50,65,145]. In contrast, Wnt8-Fzd5/8-JNK-Sp5 signaling activates *wnt8* expression in the same equatorial ectoderm cells in which the ANE GRN is downregulated, indicating that ANE restriction in these more anterior cells is driven by a Wnt8-Fzd5/8-JNK-Sp5 positive feedback mechanism [50,76] (Figure 3B–D and Figure 4A).

As discussed above, there is considerable conservation among the ANE GRN components in several animals, from cnidarians to vertebrates. In addition, these ANE GRNs are initially broadly expressed throughout the anterior ectoderm in several other invertebrate deuterostome embryos (sea stars, hemichordates, and cephalochordates) as well as the early neural plate in vertebrates (for review, see Range, 2014 [2]). Several functional studies in these deuterostomes indicate that a posterior-to-anterior Wnt/β-catenin-dependent mechanism downregulates ANE genes from the equatorial and/or posterior neural plate territories in these embryos [146,147,148,149] (Figure 4). Strengthening the idea that aspects of the ANE restriction mechanism identified in sea urchins are conserved among deuterostomes is that orthologues to components within the sea urchin ANE restriction mechanism mentioned above are similarly expressed during ANE restriction in invertebrate and vertebrate deuterostome embryos (Figure 4) (for review, see [2]). Importantly, functional studies in zebrafish embryos show that a strikingly similar posterior-to-anterior mechanism using Wnt8-Fzd8-Sp5-like is critical to position the ANE GRN to the anterior neural plate (Sp5-like is an orthologue of sea urchin Sp5) (Figure 4D) [150,151,152,153,154]. Together, these studies have led us to propose that aspects of this mechanism existed in the common deuterostome ancestor [2,50]. In the pre-bilaterian Hydra, a Wnt-TCF-Sp5-Zic4 signaling mechanism restricts the head organizer, lending further support to our argument that the Wnt/Sp5 cassettes play an ancient, conserved role in body axis formation [28,29]. Thus, it would be interesting in the future to assess whether non-canonical Wnt/JNK signaling is a conserved feature of AP patterning in phylogenetically distinct embryos.

**Figure 4 jdb-13-00036-f004:**
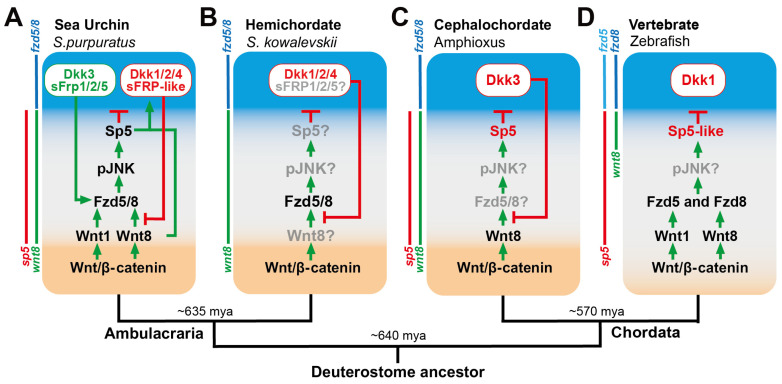
Schematic diagrams showing similarities among the ANE GRN restriction mechanisms among deuterostome embryo model systems. Embryos are shown at the approximate stages when ANE restriction has finished (early to late gastrula). A progressive posterior-to-anterior Wnt/β-catenin-dependent mechanism downregulates ANE GRN components from posterior ectodermal cells in each embryo. The components shown in light gray indicate that the function of the gene has not been assessed in that embryo. Colored bars next to each embryo indicate the spatial expression domains for the Wnt and Fzd receptors that are necessary for ANE GRN restriction in each embryo. (**A**) Sea urchin model (**B**) Hemichordate model (**C**) Cephalochordate model (**D**) Vertebrate model See text for details. Expression and functional data taken from [2,78,150,155,156,157,158,159,160].

### 2.3. An Anterior Signaling Center Defines the Boundaries of the ANE Territory

A two-gradient model for cell fate specification along the sea urchin anterior–posterior axis was proposed by two prominent sea urchin embryologists, John Runstrom and Sven Horstadius, in the mid-20th century [58]. This model was based on classical cut and paste embryological experiments before the advent of molecular biology. It states that the sea urchin embryos contain two early AP/AV gradients that influence cell fate specification, one emanating from the posterior/vegetal pole and another from the anterior/animal pole. These two gradients then interact to mutually specify cell fate and are partially ‘hostile’ to each other during this process. With the advent of molecular gene perturbations, this model fell out of favor, but we argue here that several recent studies in the sea urchin provide some support for this idea.

The cWnt to non-canonical Wnt/JNK signaling relay mechanism described above clearly shows that a posterior-to-anterior Wnt signaling center is critical to specify and pattern the germ layer GRNs along the sea urchin AP axis (Figure 2 and Figure 3), and recent studies indicate that there are several factors expressed around the anterior pole that are ‘hostile’ to the Wnt/JNK signaling. One such Wnt antagonist is the transcription factor Six3/6 mentioned above as one of the first GRN components activated in the ANE GRN (Figure 3). Gain-of-function and morpholino knockdown experiments initially suggested that Six3/6 is both necessary to activate the entire ANE GRN and antagonize posterior-to-anterior Wnt signaling at the transcriptional level [64]. However, a recent study showed that ANE GRN components are still broadly expressed in these embryos in cWnt and Six3/6 double-knockdown embryos [67]. This result indicates that the broadly expressed ‘ANE default mechanism’ can activate the ANE GRN without Six3/6, suggesting that Six3/6’s primary role in ANE specification is its antagonism of the posterior-to-anterior Wnt signaling ANE restriction mechanism.

Expression and functional analyses in vertebrate, cephalochordate, hemichordate, and echinoderm embryos show that several related anteriorly expressed secreted Wnt signaling modulators antagonize Wnt signaling emanating from the posterior/vegetal pole [2]. These studies argue for a deep homology for these Wnt modulators in primary axis formation. Of these secreted Wnt modulators, the ancestral Dkk1/2/4 and its vertebrate orthologues are arguably the most well-known and are involved in early primary axis patterning in several organisms from Cnidarians to vertebrates [1,18] (Figure 3D, Figure 4A and Figure 5). This molecule prevents Wnt ligands and the Wnt co-receptor LRP5/6 from interacting, which is essential for Wnt signaling in many cellular contexts, including ANE restriction in sea urchins [113,161,162,163] (Figure 5). In sea urchin embryos, Wnt8-Fzd5/8-JNK-Sp5 signaling activates *dkk1/2/4* expression in a territory around the anterior pole during the *later stages* of ANE restriction (late blastula/early gastrula stage). Subsequently, Dkk1/2/4 represses Wnt8-Fzd5/8-JNK signaling within the *dkk1* expression territory in a classic negative feedback loop [50,67], defining the outer boundary of the ANE territory by the mesenchyme blastula stage (Figure 3D, Figure 4A and Figure 5).

Signaling pathways are often regulated by more than one secreted modulator, so it is unsurprising that anterior Fzd5/8 signaling also activates the expression of another secreted Wnt modulator, originally named secreted Frizzled Related Protein-1 (sFRP-1) [164], around the anterior pole during later blastula stages. sFRPs comprise the largest family of Wnt modulators in the animal kingdom, and the eumetazoan ancestor genome contained both sFRP1/5 and sFRP3/4 [6,77,165]. However, this gene is novel to the sea urchin, containing a modified domain architecture that appears to have evolved from the ancestral sFRP3/4. Therefore, we now call this gene sFRP-like to avoid confusion with the ancestral sFRP1/5 gene (Figure 3D, Figure 4A and Figure 5). Functional epistasis data indicate that sFRP-like works in parallel with Dkk1/2/4 to antagonize Fzd5/8-JNK signaling, and together these proteins *allow for the expression of the ANE GRN* around the anterior pole, reinforcing Six3/6 and defining the outer boundary of the territory [112] (Figure 3D, Figure 4A and Figure 5). It is interesting that sea urchins contain an sFrp3/4 orthologue but that it is not expressed during ANE restriction (it is transcribed during later larval stages) [112]. Reminiscent of the roles for Dkk1/2/4 and sFRP-like in the sea urchin, Dkk1 and sFRP3 (FrzB) work together to repress early posterior-to-anterior Wnt signaling and define the ANE territory of the neural plate in *Xenopus* and zebrafish embryos [166,167,168] (Figure 4D). Therefore, it is tempting to speculate that sFRP-like may have become integrated into the ANE restriction mechanism in sea urchins, eventually assuming the role of the ancestral sFRP3/4 in establishing the ANE territory.

Anterior Six3/6, Dkk1/2/4, and sFRP-like all clearly act as ‘hostile’ forces against posterior-to-anterior Wnt signaling as suggested by the two-gradient model, but we have also found the ANE territory secretes Wnt modulators that also promote posterior-to-anterior Wnt signaling and the downregulation of the ANE GRN in the equatorial ectoderm. As the cardinal ANE GRN regulator FoxQ2 is progressively restricted to the anterior pole of the sea urchin, it activates two additional secreted Wnt modulators, Dkk3 and the aforementioned sFRP1/5, within the anterior-most ANE domain at approximately the same time as Dkk1/2/4 and sFRP-like around late blastula stages (Figure 3D, Figure 4A and Figure 5) [50,114]. Dkk3 and Dkk1/2/4 occupy the two ancestral branches of the Dkk family [169]. Studies indicate that while most Dkk1/2/4 orthologues antagonize Wnt signaling by interfering with the Fzd co-receptors Lrp5/6 and Kremen, it is still unclear how Dkk3 functions in the Wnt pathway [114,161,163,170,171,172,173]. Also, while sFRPs are generally considered to be Wnt antagonists, recent data indicate that they can also act as Wnt signaling agonists by promoting Wnt ligand diffusion in a concentration dependent manner [174,175,176,177]. Based on these studies, it is interesting that a series of functional epistasis experiments show that anteriorly expressed Dkk3 and low levels of sFRP1/5 counteract the antagonism of the Wnt8-Fzd5/8-JNK-Sp5 signaling by Dkk1/2/4 and sFRP-like in the sea urchin [114] (Figure 3D, Figure 4A and Figure 5). But consistent with the dual function of sFRP1/5 in other systems, Wnt8-Fzd5/8-JNK-Sp5 signaling is inhibited in embryos overexpressing high levels of sFRP1/5 mRNA (Figure 5). Currently, our model is that sFrp1/5 and Dkk3 diffuse in an anterior-to-posterior gradient from the inner ANE territory specified by FoxQ2. Then Dkk3 and low levels of sFRP1/5 positively modulate Wnt8-Fzd5/8-JNK-Sp5 signaling in cells in and around the forming ANE boundary, promoting the posterior-to-anterior Wnt1/Wnt8-Fzd58-JNK-mediated downregulation of ANE genes in these cells and counteracting the antagonistic influence of Dkk1/2/4 and sFRP-like. However, high levels of sFRP1/5 around the apex of the anterior pole work in concert with Dkk1/2/4 and sFRP-like to antagonize Wnt8-Fzd5/8-JNK signaling within the newly forming ANE territory (Figure 5). Thus, the positioning and proper sizing of the ANE territory, and by extension the equatorial ectodermal territory, depends on an elegant balancing act between posterior-to-anterior activity of Wnt1/Wnt8-Fzd5/8-JNK signaling and the opposing positive and negative influence of the four anteriorly expressed secreted Wnt signaling modulators (Figure 5).

We propose that the studies detailed above are reminiscent of the AP/AV double gradient model first put forth by Runstrum and Hörstadius. They envisioned two positive signaling centers around opposite poles of the sea urchin primary axis that secreted morphogens that were necessary to specify cell fates around each pole while counteracting each other’s influence in the equatorial regions of the early embryo. The current data support a similar but more nuanced modular model where two separate Wnt signaling modules (cWnt in posterior blastomeres and Wnt8-Fzd5/8-JNK in anterior ectoderm cells) progressively specify and pattern germ layer GRNs from the posterior to the anterior ends of the embryo during cleavage and blastula stages (Figure 5). These two Wnt signaling modules are initially connected when Wnt1 and Wnt8 diffuse from posterior EM blastomeres to activate Fzd5/8-JNK signaling in more anterior ectoderm cells around the 60-cell stage. Then Wnt8 plays a more prominent role in ANE GRN downregulation in the ectoderm during later cleavage and early blastula stages since it is activated in these cells in a Wnt8-Fzd5/8-JNK-Sp5 positive feedback loop (Wnt1 expression is restricted to posterior EM cells). Finally, the anterior signaling center acts as a third module when it begins secreting Wnt modulators during late blastula stages that regulate Wnt8-Fzd5/8-JNK-Sp5 signaling activity, helping establish the critical boundary between primarily non-neural equatorial ectoderm and the ANE. Currently it is unclear if any of the Wnt modulators of the anterior signaling center are also involved in activating genes involved in cell fate specification rather than regulating the later patterning events mediated by Wnt/JNK signaling. But recent data indicate that the territory contains at least five concentric domains of gene expression (which are also subdivided along the DV axis) and that we currently have a poor understanding of how these domains are specified and patterned. Therefore, it is possible that both Dkk3 and sFRP1/5, along with other uncharacterized secreted Wnt signaling ligands and modulators, may also positively regulate the specification of gene expression in the ANE and/or more anterior territory of the equatorial ectoderm. If so, these anteriorly expressed morphogens would then be more consistent with Runstrum and Hörstadius’s model that posits that positive morphogens from the anterior are in competition with positive morphogens emanating from the vegetal/posterior pole.

### 2.4. Canonical and Non-Canonical Frizzled1/2/7 Signaling During AP Formation

It is not uncommon for Wnt ligands and Fzd receptors to activate multiple Wnt pathways in the same cells and/or territories [88,178,179,180], but exactly how this toggling among different pathways is achieved in any given cellular context is not fully understood. One way that this switch could occur is through context dependent combinations of ligand-receptor pairs. For example, in amphibians, both Wnt5 and Wnt11 have been shown to signal through either cWnt or non-canonical Wnt/Ca^2+^ pathways depending on the Frizzled receptors with which they are paired [88,180,181,182,183,184,185,186]. This is also the case for the Fzd7 receptor in *Xenopus*, which, when paired with the Wnt8 ligand, activates β-catenin signaling during DV patterning, but when paired with Wnt5 and/or Wnt11, Fzd7 activates JNK and Protein Kinase C (PKC) as second messengers, driving convergent extension during gastrulation in the same cells [180,187,188,189,190].

A similar phenomenon has been observed in sea urchins, where the same Fzd orthologue, Fzd1/2/7, signals through both canonical and non-canonical Wnt pathways depending on its ligand pair during AP formation. The first evidence for a dual function for Fzd1/2/7 in sea urchin AP formation came from a study attempting to determine the possible roles of maternal Wnt components in the early initiation of Wnt/β-catenin signaling. In this study, Croce et al. 2011 conducted morpholino knockdowns of the three maternally supplied and broadly expressed Wnts (Wnt6, Wnt7, and Wnt16), but none of these knockdowns perturbed the initiation of cWnt signaling as expected [75]. However, they did find that Wnt6 knockdown embryos failed to activate several endoderm GRN genes (*wnt8*, *foxa*, *brachyury*, and *blimp1*) during blastula stages while leaving mesoderm formation unperturbed. They also found that perturbing the function of the broadly expressed maternal Fzd1/2/7 receptor phenocopied these results and that Wnt6 and Fzd1/2/7 likely function as a ligand–receptor pair in macromeres. Finally, they discovered that antibody stains for β-catenin in the Fzd1/2/*7* perturbed embryos showed reduced nβ-catenin in the posterior macromeres whose progeny form endoderm and non-skeletogenic mesoderm cells, but not micromeres, which will give rise to the larval skeleton. Together, these data indicate that in posterior blastomeres, Wnt6-Fzd1/2/7 signaling maintains nβ-catenin in macromeres during early cleavage stages, which is critical for the regulation of several endoderm genes, but that Wnt6-Fzd1/2/7 signaling does not initiate nβ-catenin in posterior blastomeres, which is consistent with subsequent evidence that strongly suggests that nuclear localization of β-catenin around the 16-cell stage is ligand independent (Figure 5) [49].

From here the story becomes more complicated because during the same temporal window that Wnt6-Fzd1/2/7 signaling is promoting cWnt signaling in posterior/vegetal blastomeres, subsequent studies show that Fzd1/2/7 signaling using a different ligand activates non-canonical Wnt signaling in the same blastomeres [50,53]. In these studies, Fzd1/2/7 morpholino perturbations showed that the entire ANE GRN is eliminated as early as the 32-cell stage. This phenotype is the opposite of the cWnt and Wnt/JNK knockdown phenotypes that show broad ectopic ANE GRN expression. These data indicate that Fzd1/2/7 signaling is broadly active throughout the embryo and that it transduces its signal through a different non-canonical Wnt pathway than Wnt/JNK, most likely the Wnt/Ca^2+^ pathway. Supporting this idea is that embryos treated with a pan-phosphorylation inhibitor of Protein Kinase C (PKC) proteins, which are often involved in Wnt/Ca^2+^ signaling, phenocopied Fzd1/2/7 knockdown embryos, and PKC was not phosphorylated in the absence of Fzd1/2/7 signaling [50]. In addition, epistasis experiments in which Fzd1/2/7 morphants or embryos treated with a pan-PKC inhibitor were co-injected with dominant negative Fzd5/8 rescued *foxq2* expression. Both experiments suggest that Fzd1/2/7-PKC antagonizes Wnt8-Fzd5/8-JNK signaling in anterior blastomeres, and Wnt16 has been identified as the ligand necessary to activate the Fzd1/2/7-PKC pathway mediated antagonism of Wnt8-Fzd5/8-JNK signaling in the equatorial ectoderm [53]. However, while the reporter gene assay for cWnt signaling (TOP-FLASH) showed elevated posterior cWnt signaling in the absence of Fzd1/2/7 signaling, it was unaffected in PKC-inhibited embryos [50]. This result indicates that Fzd1/2/7 signaling in posterior blastomeres antagonizes cWnt signaling independent of PKC, possibly through another component of the Wnt/Ca^2+^ pathway (Figure 5). It is also unclear whether Wnt16 is involved in antagonism in the antagonism of cWnt signaling in posterior/vegetal blastomeres.

Taken together, these results strongly suggest that Fzd1/2/7 is necessary for at least three different, but interconnected, early AP mechanisms in sea urchins: (1) Wnt6-Fzd1/2/7 signaling maintains nβ-catenin in posterior blastomeres during early cleavage and blastula stages, regulating aspects of the endoderm GRN; (2) in the same posterior blastomeres, Fzd1/2/7 signaling activates an as of yet uncharacterized non-canonical intracellular transduction mechanism that antagonizes posterior cWnt signaling; and (3) Wnt16-Fzd1/2/7-pPKC signaling antagonizes the Wnt1/Wnt8-Fzdl5/8-JNK-Sp5-mediated ANE restriction mechanism in anterior blastomeres (Figure 5). Importantly, in the absence of Fzd1/2/7 signaling, the ANE GRN is displaced by equatorial ectoderm genes around the anterior pole, and the endomesoderm territory expands into the equatorial territory by the mesenchyme blastula stage in Fzd1/2/7 morphants [50,67]. This result indicates that these three roles for Fzd1/2/7 signaling are integrated to globally regulate the positions of the primary germ layers along the AP axis by the beginning of gastrulation. Thus, while the Wnt/Ca^2+^ pathway has not been fully characterized in the sea urchin embryos, it will be important to investigate the role of putative Ca^2+^ signaling components (i.e., calmodulin, IP3, and CAMKII) as well as other extracellular modulators to clarify how a single Wnt receptor modulates several intracellular responses, even in the same cells.

### 2.5. Perspectives

Understanding how the myriad of extracellular and intracellular signaling components interact to form Wnt signaling networks is not a trivial problem to solve in any cellular context. It is a fascinating puzzle that many scientists in the Wnt signaling field have been attempting to understand for the last 15 years, once it became evident that Wnt signaling pathways do not generally exist as isolated, linear pathways. Instead, multiple studies like those detailed here have shown that different Wnt pathways often interact with one another in unpredictable ways within the same cells/territories [4,44]. The dynamic four dimensional spatial and temporal expression changes in specific Wnt signaling components that occur in developmental processes make it especially challenging to determine specific Wnt signaling interactions among cells/territories.

Here we would like to emphasize that the right model is critical to assess how Wnt signaling interactions control any specific developmental process. As mentioned above, most of our understanding of AP specification and patterning comes from vertebrate, fly, and worm studies. Based on comparative expression and functional studies in many animals, it appears that both flies and worms use derived AP patterning mechanisms, whereas vertebrate embryos use the ancestral Wnt mechanism. But vertebrate Wnt-mediated AP germ layer patterning occurs during the complicated morphogenetic movements of gastrulation, which is largely controlled by non-canonical Wnt/JNK and Wnt/Ca^2+^ pathways. Therefore, perturbation of these non-canonical pathways causes morphogenetic defects that may obscure any patterning roles for these pathways since it is difficult to identify embryonic territories by marker gene expression in these perturbed embryos.

In a stroke of evolutionary luck, the early specification and final position of the germ layer GRNs are established by the sea urchin AP Wnt network in a non-motile single cell thick epithelium before gastrulation. This difference in the developmental timing of early AP patterning in comparison to vertebrates has allowed us to assess the transcriptional role of Wnt/JNK and Wnt/PKC signaling during this process, making it possible to uncover the integrated canonical and non-canonical AP Wnt signaling network in sea urchins. It is possible that sea urchins integrated the Wnt/JNK and Wnt/PKC signaling pathways into the ancestral cWnt AP patterning mechanism during their evolution from the last common deuterostome ancestor. But as we discussed above, functional and expression studies indicate that there is a remarkable conservation among the Wnt signaling components that drive AP formation among vertebrates, sea urchins, and other animals. Therefore, it is tempting to speculate that instead of a sea urchin novelty, aspects of the AP Wnt network described here may be shared broadly among many animals, including non-canonical Wnt signaling.

In the future it will be important to perform functional studies in a diverse group of metazoan organisms to assess the conservation of the sea urchin AP Wnt network. These studies could shed light on the origins and the evolution of this ancient, fundamental axial patterning mechanism. We also think it is important to note that Wnt signaling pathways appear to be deeply conserved throughout metazoans. This is an important consideration because the interconnectivity among the many components involved in the different Wnt signaling pathways may be hardwired during AP axis formation (and in other conserved developmental processes). Thus, understanding the Wnt signaling network interactions in diverse model systems may be instrumental in uncovering the mechanisms underlying general aspects of how Wnt signaling pathways interact in any cellular context.

## Figures and Tables

**Figure 3 jdb-13-00036-f003:**
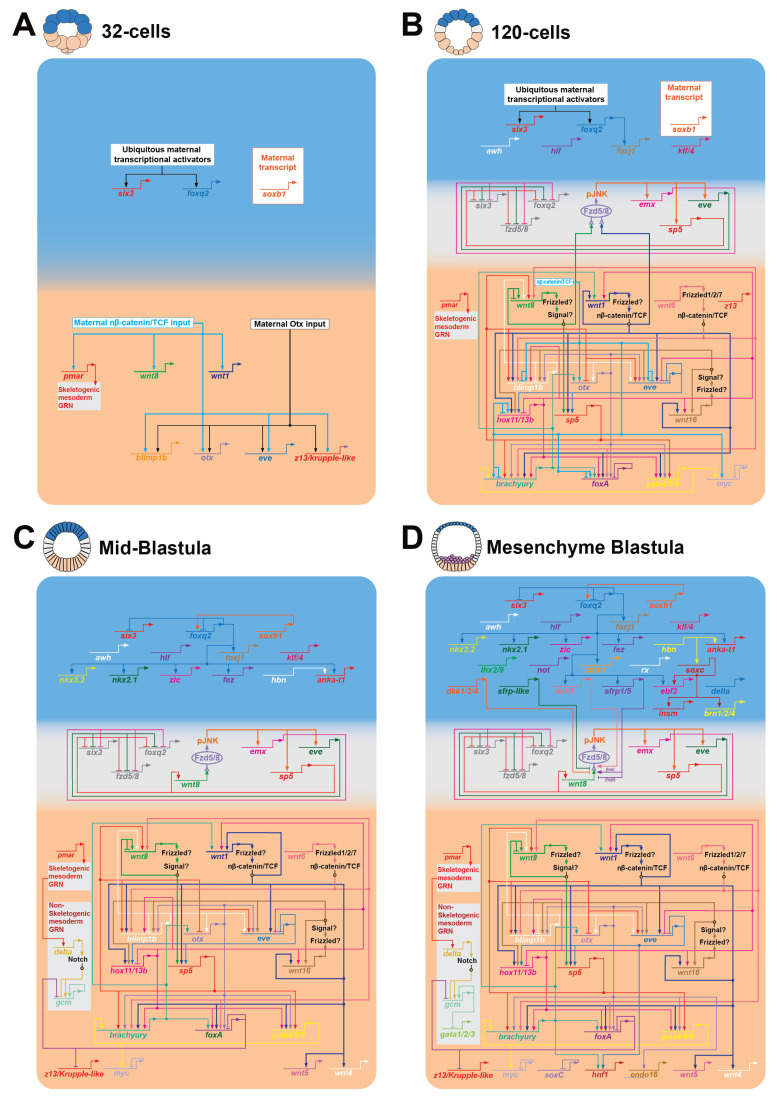
The progressive deployment of the endomesoderm, equatorial ectoderm, and anterior neuroectoderm gene regulatory networks from cleavage to early gastrula stages. (**A**) At the 32-cell stage, ubiquitous maternal factors activate the ANE GRN in the anterior half of the embryo, while localized maternal intracellular cWnt components nuclearize β-catenin, activating the EM GRN in the posterior half of the embryo. (**B**) As early as the 60-cell stage, mid-level regulatory genes are activated in the ANE and EM GRNs, and Wnt1/Wnt8-Fzd5/8-JNK signaling downregulates the ANE GRN from the equatorial ectoderm. (**C**) By the mid-blastula stage, the endoderm and mesoderm territories begin to segregate, and mid-level regulatory genes are activated in the ANE GRN. (**D**) At the mesenchyme blastula stage, signaling feedback loops between the ANE GRN and the equatorial ectoderm help define the final position of the ANE territory, while the endoderm and mesoderm GRNs are further refined to prepare for gastrulation. Data taken from [2,50,53,64,65,68,76,94,101,112,113,114,115,116,117,118,119,120,121,122,123,124,125,126,127,128]. Lateral embryo images in the figure created in BioRender (https://BioRender.com/7bogzo4, accessed on 6 July 2025).

**Figure 5 jdb-13-00036-f005:**
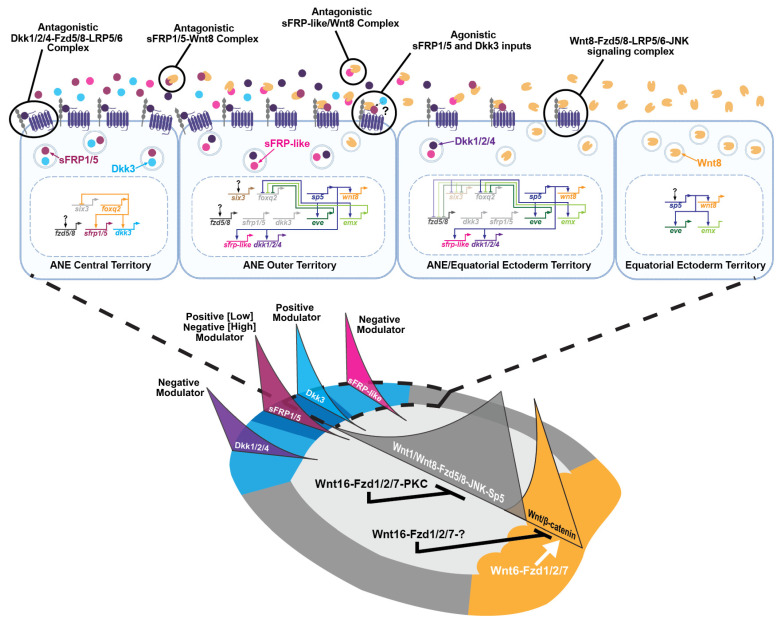
A double gradient model for early AP formation in sea urchin embryos. (**Top diagram**) A model representing interactions among the Wnt modulators, receptors, and co-receptors and their influence on emerging ANE and equatorial ectoderm GRNs around the anterior pole. (**Bottom diagram**) Overview diagram of how the Wnt/β-catenin, Wnt/JNK, and Wnt/PKC (Ca^2+^) pathways are integrated in an anterior–posterior Wnt signaling network in sea urchin embryos. Data taken from [50,53,67,76,112]. The top diagram was created in BioRender on (https://BioRender.com/o3nvkt4, accessed on 6 July 2025).

## Data Availability

No new data was produced in this review.

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
