# Peer review of "An Integrated Canonical and Non-Canonical Wnt Signaling Network Controls Early Anterior–Posterior Axis Formation in Sea Urchin Embryos"

_jdb, 2025, doi:10.3390/jdb13040036_

Round 1
Reviewer 1 Report
Comments and Suggestions for Authors
The authors delineate the molecular signaling events involving Wnt signaling that give rise to the sea urchin main AP axis. They convincingly argue that Wnt signaling forms an interconnected network of interactions involving multiple players at different times, rather than separated canonical and non-canonical cascades. In addition, the spatial organization of Wnt signaling appears to correspond to the classic double-gradient ideas of Runnström and Hörstadius.
The review is interesting and provides a sufficiently comprehensive summary of the current state of knowledge in the sea urchin field. The concept of interconnectivity within the Wnt cascades, while not entirely new, is supported with a concrete example that merits further dissemination. Overall, I am supportive of the publication of this manuscript, but I recommend several modifications that would strengthen it considerably:
- Evolutionary discussion: The section on evolution contains some omissions. While the authors briefly mention cnidarians and Nematostella, they omit a more detailed discussion of Wnt signaling in Hydra. This is particularly important given recent findings on the role of Sp5 in Hydra axis regulation.
- Wnt signaling in sponges: It is even more important to include a discussion of Wnt signaling in sponges. Such a comparison may help clarify whether Hox genes or Wnt signaling represent the most ancient mechanism of axial patterning (see lines 74–78).
- Double-gradient concept: The revival of the double-gradient idea is certainly intriguing. However, such concepts can also be limiting if they become rigid frameworks into which all findings are forced. For the sake of balance, it would be worthwhile to mention possible limitations of this view—for example, that what appears as two gradients is in fact only a single gradient.
- Implications for morphogen models: An especially thought-provoking consequence of the authors’ model of Wnt signaling in sea urchin embryos is that traditional morphogen-gradient tenets may become superfluous. The model suggests that positional information does not depend on reading morphogen concentrations, but rather on a zonation of distinct Wnt signaling modalities. This point could be emphasized more explicitly, as it highlights an important conceptual shift.
- Figure legends: The figure legends could be more complete and consistent. At present, some legends provide detailed descriptions of panels while others do not. In Figure 2, panels D and E are not mentioned at all. Revising the legends for completeness and homogeneity would improve clarity for readers.
Recommendation:
I recommend acceptance of this review after minor revisions to address the points above. These adjustments will enhance both the scholarly depth and readability of the manuscript without requiring substantial restructuring.
Author Response
Reviewer 1
Comments and Suggestions for Authors
The authors delineate the molecular signaling events involving Wnt signaling that give rise to the sea urchin main AP axis. They convincingly argue that Wnt signaling forms an interconnected network of interactions involving multiple players at different times, rather than separated canonical and non-canonical cascades. In addition, the spatial organization of Wnt signaling appears to correspond to the classic double-gradient ideas of Runnström and Hörstadius.
The review is interesting and provides a sufficiently comprehensive summary of the current state of knowledge in the sea urchin field. The concept of interconnectivity within the Wnt cascades, while not entirely new, is supported with a concrete example that merits further dissemination. Overall, I am supportive of the publication of this manuscript, but I recommend several modifications that would strengthen it considerably:
- Evolutionary discussion: The section on evolution contains some omissions. While the authors briefly mention cnidarians and Nematostella, they omit a more detailed discussion of Wnt signaling in Hydra. This is particularly important given recent findings on the role of Sp5 in Hydra axis regulation.
We thank the reviewer for pointing out this omission and have now added to the text both in the introduction and in the later section (title: A cWnt to Wnt/JNK signaling relay mechanism positions ectodermal GRNs along the AP axis) where we discuss the role of the Wnt/Sp5 cassette in the sea urchin. We feel that this addition is important to further highlight the deep conservation among the AP Wnt components across metazoans.
- Wnt signaling in sponges: It is even more important to include a discussion of Wnt signaling in sponges. Such a comparison may help clarify whether Hox genes or Wnt signaling represent the most ancient mechanism of axial patterning (see lines 74–78).
We have modified the text to include the information that Hox genes were once considered absent in sponges and ctenophores because they were lacking in the Amphimedon and Pleurobrachia genomes. However, Hox and Parahox genes have been found to be present and expressed during development in both sponges and ctenophores (calcareous sponges and Mnemiopsis). Thus, it is still unclear whether Wnt or Hox is the more ancient primary axis patterning mechanism.
- Double-gradient concept: The revival of the double-gradient idea is certainly intriguing. However, such concepts can also be limiting if they become rigid frameworks into which all findings are forced. For the sake of balance, it would be worthwhile to mention possible limitations of this view—for example, that what appears as two gradients is in fact only a single gradient.
See our comments in critique 4 where we address both concerns since they are related.
- Implications for morphogen models: An especially thought-provoking consequence of the authors’ model of Wnt signaling in sea urchin embryos is that traditional morphogen-gradient tenets may become superfluous. The model suggests that positional information does not depend on reading morphogen concentrations, but rather on a zonation of distinct Wnt signaling modalities. This point could be emphasized more explicitly, as it highlights an important conceptual shift.
We agree that the double gradient model is a somewhat rigid framework since it is based on two simple ‘positive’ morphogen gradients that compete against one another to specify and pattern embryonic territories. We decided to use the Rundstrom and Hörstadius model to highlight our more nuanced AP specification and patterning model in sea urchins that relies on discrete Wnt signaling regulatory modules that are activated by different Wnt intracellular transduction pathways. This gets to the question above. Based on the available data it is difficult for us to see how cells are interpreting a single gradient along the AP axis since specification and patterning relies on a relay mechanism involving two different signaling pathways - posterior cWnt gradient (there is no evidence for cWnt in the ectoderm) and Wnt8-Fzd5/8-JNK-SP5 signaling in the equatorial ectoderm. Once cWnt signaling hands off the role of ANE GRN downregulation to Wnt/JNK several Wnt ligands are used to to refine the EM GRN, likely through cWnt signaling, and outside of Wnt1 and Wnt8, currently none of these have been shown to be involved in ANE restriction. Therefore, we think that the reviewer’s idea of distinct Wnt signaling modalities is a better way of thinking about sea urchin AP formation and have tried to address this idea in our new iteration of the review.
Figure legends: The figure legends could be more complete and consistent. At present, some legends provide detailed descriptions of panels while others do not. In Figure 2, panels D and E are not mentioned at all. Revising the legends for completeness and homogeneity would improve clarity for readers.
We have amended the legends for figures 2 and 3 to improve clarity and provide more details where information was lacking.
Reviewer 2 Report
Comments and Suggestions for Authors
This is an important and comprehensive review on Wnt signaling pathways in the developmental context. The authors documented the recent advances in Wnt pathways that revealed how both canonical and non-canonical Wnt signaling pathways cross-regulate the A-P axis formation and prevent anterior neuroectoderm. It’s intriguing that even the same cells can utilize simultaneously different Wnt signaling pathways. I especially appreciated the discussion from an evolutionary perspective of AP Wnt signaling network in various vertebrate and invertebrate organisms. This review will serve an important resource for the scientific community.
Minor comments:
- In the abstract, line 19, I suggest to exchange the word, ‘argue’.
- In the introduction, line 44, ‘Drosophila’ should be italicized.
- In the introduction, lines 79-81, references for this sentence should be included.
- In the introduction, lines104-106, please add references for this sentence.
- 1 caption “Frizzled” is misspelled.
- Line 152, please reference the original work in addition to the reviews in references 42 and 44.
- In section with lines 226-241, comment on which Wnt pathways ROR1/2 and Ryk are associate with.
- Line 248, “…. Protein structures among Wnts as well as Frizzleds are” Grammar needs to be fixed.
- Nomenclature on Frizzled is not consistent: the authors used “Fd” (Fig. 2 caption), “FZD”, “Frizzle”, “FZL” (line 473) throughout the review. I would use one.
- Add commas to this sentence in line 352-353: “Using zebrafish, the Weinburg lab performed a similar study, but in addition to knocking down BMP2,4,7 and ADMP, they also knocked down both zebrafish β-catenin paralogues.”
- Add comma to line 376: “…if cells have nuclearized β-catenin, they will be specified…”
- For Fig. 5, use inclusive color schematics (https://pmc.ncbi.nlm.nih.gov/articles/PMC7040535/) and avoid green and red.
- Line 538: clarify if this would be an indirect regulation.
- Line 566: “Fzd1/2/7” missing a forward slash.
Author Response
Reviewer 2
Comments and Suggestions for Authors
This is an important and comprehensive review on Wnt signaling pathways in the developmental context. The authors documented the recent advances in Wnt pathways that revealed how both canonical and non-canonical Wnt signaling pathways cross-regulate the A-P axis formation and prevent anterior neuroectoderm. It’s intriguing that even the same cells can utilize simultaneously different Wnt signaling pathways. I especially appreciated the discussion from an evolutionary perspective of AP Wnt signaling network in various vertebrate and invertebrate organisms. This review will serve an important resource for the scientific community.
Minor comments:
- In the abstract, line 19, I suggest to exchange the word, ‘argue’. Changed the wording to remove argue.
- In the introduction, line 44, ‘Drosophila’ should be italicized. Corrected.
- In the introduction, lines 79-81, references for this sentence should be included. Citations added.
- In the introduction, lines104-106, please add references for this sentence. References for this sentence were placed a line below. Citations have been moved to the correct position.
- 1 caption “Frizzled” is misspelled. Corrected.
- Line 152, please reference the original work in addition to the reviews in references 42 and 44. Added original references along with review references.
- In section with lines 226-241, comment on which Wnt pathways ROR1/2 and Ryk are associate with. Added text and relevant references explaining that across multiple model systems (Xenopus, zebrafish, C.elegans) Ror and Ryk have been shown to interact with both canonical and non-canonical pathways depending on cellular context.
- Line 248, “…. Protein structures among Wnts as well as Frizzleds are” Grammar needs to be fixed. Corrected.
- Nomenclature on Frizzled is not consistent: the authors used “Fd” (Fig. 2 caption), “FZD”, “Frizzle”, “FZL” (line 473) throughout the review. I would use one. Corrected inconsistent nomenclature.
- Add commas to this sentence in line 352-353: “Using zebrafish, the Weinburg lab performed a similar study, but in addition to knocking down BMP2,4,7 and ADMP, they also knocked down both zebrafish β-catenin paralogues.” Corrected.
- Add comma to line 376: “…if cells have nuclearized β-catenin, they will be specified…” Corrected.
- For Fig. 5, use inclusive color schematics (https://pmc.ncbi.nlm.nih.gov/articles/PMC7040535/) and avoid green and red. We have replaced all green and red colors in an effort to use more inclusive color schematics. But given the complexity of the figure there is a need for a variety of colors in order to create enough contrast to differentiate between the various molecules.
- Line 538: clarify if this would be an indirect regulation. We have amended the text in this paragraph to clarify the series of regulatory events that occur during sea urchin axial patterning.
- Line 566: “Fzd1/2/7” missing a forward slash. Corrected.
Reviewer 3 Report
Comments and Suggestions for Authors
In this manuscript, Fenner et al review what is known about the role of the Wnt signaling pathways in regulating the specification and patterning of the AV/AP axis in sea urchins. Some details of this process have been exquisitely molecularly dissected in the sea urchin, with significant contributions coming from the Fenner/Range group. The simplicity of the sea urchin embryos has provided a useful system to study these key events in early embryonic patterning and the authors do an excellent job of highlighting the similarities between the echinoderm system and other ambulacrarians and chordates. Based on many years of data in the sea urchin system the authors point out that the molecular gradients of Wnt activation and Wnt inhibition along the AV/AP axis is strikingly similar to the gradients proposed by Runstrom and Horstadius over a 100 years ago based purely on embryological data. Overall, this is a very comprehensive and well written review and it will be a very useful reference for developmental biologists interested in the specification, patterning and evolution of the primary embryonic axis in animal embryos. I have no major criticisms of the manuscript. Some minor edits are suggested below.
Minor edits:
Ref 27 is incorrect should be Wikramanayake et al 2003 (doi: 10.1038/nature02113)
Line 127, APC is repeated. One should be replaced with “Axin”
Line 159, delete (Fig. 2 Aa)
In line 183, 9 should be replaced by Nine at the start of the sentence.
In line 303 unclear why fig 3 is referenced for this statement
In line 310 “top of the” is repeated
Author Response
Reviewer 3
Comments and Suggestions for Authors
In this manuscript, Fenner et al review what is known about the role of the Wnt signaling pathways in regulating the specification and patterning of the AV/AP axis in sea urchins. Some details of this process have been exquisitely molecularly dissected in the sea urchin, with significant contributions coming from the Fenner/Range group. The simplicity of the sea urchin embryos has provided a useful system to study these key events in early embryonic patterning and the authors do an excellent job of highlighting the similarities between the echinoderm system and other ambulacrarians and chordates. Based on many years of data in the sea urchin system the authors point out that the molecular gradients of Wnt activation and Wnt inhibition along the AV/AP axis is strikingly similar to the gradients proposed by Runstrom and Horstadius over a 100 years ago based purely on embryological data. Overall, this is a very comprehensive and well written review and it will be a very useful reference for developmental biologists interested in the specification, patterning and evolution of the primary embryonic axis in animal embryos. I have no major criticisms of the manuscript. Some minor edits are suggested below.
Minor edits:
Ref 27 is incorrect should be Wikramanayake et al 2003 (doi: 10.1038/nature02113) Corrected.
Line 127, APC is repeated. One should be replaced with “Axin” Corrected.
Line 159, delete (Fig. 2 Aa) Corrected.
In line 183, 9 should be replaced by Nine at the start of the sentence. Corrected.
In line 303 unclear why fig 3 is referenced for this statement Corrected to reference relevant Fig. 4.
In line 310 “top of the” is repeated Corrected.